# Exploring the Dynamic Changes of Brain Lipids, Lipid Rafts, and Lipid Droplets in Aging and Alzheimer’s Disease

**DOI:** 10.3390/biom14111362

**Published:** 2024-10-26

**Authors:** Michele Cerasuolo, Irene Di Meo, Maria Chiara Auriemma, Giuseppe Paolisso, Michele Papa, Maria Rosaria Rizzo

**Affiliations:** 1Department of Advanced Medical and Surgical Sciences, University of Campania “Luigi Vanvitelli”, 80138 Naples, Italy; michele.cerasuolo@studenti.unicampania.it (M.C.); irene.dimeo@unicampania.it (I.D.M.); mariachiara.auriemma1@studenti.unicampania.it (M.C.A.); giuseppe.paolisso@unicampania.it (G.P.); 2Laboratory of Neuronal Networks Morphology and System Biology, Department of Mental and Physical Health and Preventive Medicine, University of Campania “Luigi Vanvitelli”, 80138 Naples, Italy; michele.papa@unicampania.it

**Keywords:** brain lipids, lipid rafts, lipid droplets, aging, Alzheimer’s disease

## Abstract

Aging induces complex changes in the lipid profiles across different areas of the brain. These changes can affect the function of brain cells and may contribute to neurodegenerative diseases such as Alzheimer’s disease. Research shows that while the overall lipid profile in the human brain remains quite steady throughout adulthood, specific changes occur with age, especially after the age of 50. These changes include a slow decline in total lipid content and shifts in the composition of fatty acids, particularly in glycerophospholipids and cholesterol levels, which can vary depending on the brain region. Lipid rafts play a crucial role in maintaining membrane integrity and facilitating cellular signaling. In the context of Alzheimer’s disease, changes in the composition of lipid rafts have been associated with the development of the disease. For example, alterations in lipid raft composition can lead to increased accumulation of amyloid β (Aβ) peptides, contributing to neurotoxic effects. Lipid droplets store neutral lipids and are key for cellular energy metabolism. As organisms age, the dynamics of lipid droplets in the brain change, with evidence suggesting a decline in metabolic activity over time. This reduced activity may lead to an imbalance in lipid synthesis and mobilization, contributing to neurodegenerative processes. In model organisms like Drosophila, studies have shown that lipid metabolism in the brain can be influenced by diet and insulin signaling pathways, crucial for maintaining metabolic balance. The interplay between lipid metabolism, oxidative stress, and inflammation is critical in the context of aging and Alzheimer’s disease. Lipid peroxidation, a consequence of oxidative stress, can lead to the formation of reactive aldehydes that further damage neurons. Inflammatory processes can also disrupt lipid metabolism, contributing to the pathology of AD. Consequently, the accumulation of oxidized lipids can affect lipid raft integrity, influencing signaling pathways involved in neuronal survival and function.

## 1. Introduction

The aging process induces changes across all biological levels, prompting the body to develop mechanisms for balance [1]. Unlike uniform decay, the human brain selectively damages specific vulnerable neuronal populations during aging [2]. Changes in the aging brain include dendrite loss, activation of microglia and astrocytes, alterations in neuronal synapses, modifications in ependymal cells, and cytoskeletal changes, all contributing to deterioration in brain morphology and function [3]. These morphological changes align with the progressive nature of aging [4]. Furthermore, after the age of 50, there is a decrease in the concentrations of most lipid species in the human brain [5]. This decrease is accompanied by a reduction in the total myelin content and lipids of the nerve cell membrane [6].

Recent studies on lipid composition in different brain regions confirm these age-related alterations [7]. Studies reported the first data linking aging to changes in the lipid profile of the human brain examined the entire brain tissue without distinguishing between grey and white matter, which could have hidden regional differences [4]. Subsequent research confirmed these regional lipid changes, highlighting a reduction in cholesterol in specific brain areas [8]. Cholesterol is a lipid that undergoes significant changes as the brain ages, diminishing its levels across different brain areas [9]. This decline also influences the production and concentration of neurosteroids like pregnenolone and dehydroepiandrosterone [10].

Apart from cholesterol, other classes of lipids also undergo selective reduction with brain aging [8,11]. For instance, during aging, phospholipids show a decrease in several brain regions, as well as also polyunsaturated fatty acids (PUFAs) resulting in a reduction in arachidonic acid and adrenic acid [12].

In nerve cells, the reduction in cholesterol and other classes of lipids can have significant consequences from a structural and functional perspective. Changes in cholesterol during aging can modify cell membrane fluidity, making neuronal membranes stiff and altering their physicochemical properties [13]. Consequently, changes in cholesterol levels can lead to structural modifications that affect the function of membrane-bound proteins, disrupting signal transmission processes. This is especially important in lipid rafts, which serve as key sites where signaling proteins gather in structures known as signalosomes. Altered protein–protein interactions and rearrangements of protein complexes have been observed due to changes in cholesterol and other lipids in these rafts, causing disruptions in multiple signaling pathways that regulate neuronal function [14,15,16].

Changes in lipid metabolism can affect the homeostasis of lipid droplets, an important organelle in the nervous system present in all types of brain cells under different conditions. The formation of lipid droplets significantly affects cellular health, especially during the progression of brain diseases. Although lipid droplets are commonly found in various neuropathologies, their active role in regulating cellular stress in the brain has been appreciated only recently [17]. Research has demonstrated that changes in lipid metabolism, affecting both lipid rafts and lipid droplets, are present in neurological diseases, including neurodegenerative disorders like Alzheimer’s disease (AD). In the early stages of AD, changes in fatty acids have been reported, particularly within lipid rafts, along with increased cerebral lipid peroxidation [18]. It is known that AD is characterized by the presence of insoluble aggregates of amyloid β (Aβ), forming neuritic plaques (NPs), which play a central role in the disease’s pathology. However, the composition and arrangement of cholesterol in the cell membrane have been identified as crucial factors influencing the generation and subsequent buildup of Aβ peptides in the plasma membrane, contributing to cellular dysfunction and neuronal death [19]. Following the formation of amyloid plaques and neurofibrillary tangles, lipid droplet accumulation has also emerged as a crucial factor in AD pathology, especially observed in models with different genetic backgrounds [20]. In summary, current data show a strong association between lipid homeostasis, metabolism, and human brain aging. Considering the various functions carried out by the human brain that often decline with age, it is reasonable to suggest that alterations in the lipid composition of specific brain regions could play a role in the pathophysiological mechanisms underlying neurodegenerative disorders like Alzheimer’s disease [4,18].

This study aims to explore the intricate relationship between lipid metabolism, lipid raft changes, and lipid droplet accumulation in the aging process and AD.

Understanding the metabolic aspects of this complex disease is crucial, and this review hopes to provide insights into them.

## 2. The Physiological Role of Lipids in the Brain

Lipids are organic molecules that are soluble in nonpolar organic solvents but have poor solubility in water. Lipids primarily function as energy storage molecules and form a broad group of organic compounds that can be classified into different categories. This diversity stems from variations in the head group structure, carbon–carbon bond types, molecular weight, and overall composition. The core structure of lipids is defined by the polarity differences between their hydrophobic tails and hydrophilic head groups. These polarity variations influence lipid interactions and their roles in biological membranes [21]. Lipids play an essential role in metabolic functions and can be classified into eight categories using biochemical methods: fatty acids, glycerolipids, glycerophospholipids, sphingolipids, sterols, prenols, saccharolipids and polyketides [22].

Lipid rafts are small, dynamic membrane domains rich in cholesterol, sphingolipids, and saturated FAs and low in PUFAs [4]. Traditionally seen as mobile cholesterol-rich domains regulating signal transduction and protein sorting [23], recent studies suggest that while some rafts are mobile, others are anchored to the cytoskeleton [24]. These anchored rafts provide structural support, stabilize membranes, and are crucial for cell adhesion, lipid/protein sorting, and signal transduction [4].

Lipid droplets are active intracellular organelles in the brain that store neutral lipids like cholesteryl esters and triacylglycerols. These droplets often accumulate in specific cell types and can appear in neurons due to aging or stress [17,25,26]. Their metabolism in glia, neurons, and neural stem cells has been studied in both health and disease, though the exact mechanisms remain an area of ongoing interest [25]. Apart from adipose tissue, the brain is one of the richest sites in lipids. In fact, lipids make up 50–60% of the cell membrane constituents in nerve cells [4].

Below, we will discuss the most common types of lipids found in the brain such as glycerolipids, glycerophospholipids, sphingolipids, cholesterol, and cholesteryl esters [18,22]. However, beyond the major lipid classes in the brain, there are numerous others, and their roles are still under study and yet to be fully defined [27,28].

### 2.1. Classification and Functions

**Glycerolipids and Fatty Acids.** Triglycerides (TAGs) are the most common type of glycerolipids. TAGs represent the storage form of fatty acids (FAs), which are involved in beta-oxidation, a well-known process of breaking down fatty acids into substrates used in mitochondrial ATP production [29], in the brain and in all cells.

FAs are crucial components of neuronal membranes, present as phospholipids and cholesterol esters. Synthesized by the liver, fatty acids reach the brain via the bloodstream, crossing the blood–brain barrier (BBB) with albumin, lipoproteins, and other transporters [30]. The exact mechanisms of fatty acid transport across the BBB remain unclear, with hypotheses suggesting passive diffusion or transport mediated by specific fatty acid transport proteins [10,11]. Long-chain polyunsaturated fatty acids (LC-PUFAs), like docosahexaenoic acid (DHA, n-3 PUFAs) and arachidonic acid (AA, n-6 PUFAs), comprise around 25–30% of all fatty acids [31]. FAs can be classified into saturated and unsaturated types based on the number of double bonds they have. The major categories of FAs include saturated, trans, mono-unsaturated and polyunsaturated FAs. Saturated fatty acids (SFAs) do not have any double bonds, while unsaturated fatty acids contain at least one double bond (known as monounsaturated fatty acids or MUFA) or two or more double bonds (known as poly-unsaturated fatty acids or PUFAs) [32]. PUFAs are essential for brain function, comprising about 50% of neuronal membranes and 70% of myelin sheaths [4]. They maintain membrane integrity, influence signal transduction and gene transcription, and protect neurons from apoptosis [30]. Additionally, PUFAs are crucial for producing lipid mediators in inflammation, with n-6 fatty acids acting as precursors for eicosanoids, such as prostaglandins and leukotrienes [33]. Other glycerolipids include monoacylglycerol (MAG) and diacylglycerol (DAG) [34,35].

**Glycerophospholipids.** Glycerophospholipids are the main components of cell membranes. These lipids account for around 4.5–5% of the brain’s total wet weight and 4.5% of the grey matter [4].

Ethanolamine phosphoglyceride is the most abundant phospholipid in the human brain, making up 35.6% of its phospholipid content, with ethanolamine plasmalogen (PlsEtns) being the most prevalent form. Phosphatidylethanolamine (PE) contributes to the brain’s phospholipid composition, while phosphatidylcholine (PC) is the primary phosphoglyceride form of choline. Neural membranes contain various glycerophospholipids (32.8%) [36]. The composition of glycerophospholipids is crucial for the functional efficacy of neural membranes, as they turnover at different rates based on their structure and location [18]. The length and saturation of glycerophospholipid acyl chains, including domains enriched in polyunsaturated fatty acids, define membrane characteristics. Phospholipases A(1), A(2), C, and D degrade glycerophospholipids, producing second messengers like arachidonic acid and diacylglycerol. These messengers are crucial for apoptosis, transporter regulation, and membrane enzyme modulation, highlighting the importance of neural membrane phospholipids as reservoirs for second messengers [37].

**Sphingolipids.** Sphingolipids (SPs) are a diverse group of molecules that play crucial roles in various physiological processes. There are several hundred types of SPs, with each one having a specific function. For example, metabolic products of SPs, like ceramide, sphingosine (Sph), Sph-1-phosphate (S1P), and Cer-1-phosphate (C1P), are bioactive molecules involved in cellular processes such as signal transduction, protein sorting, cell-to-cell interactions, and recognition [38,39]. Additionally, SPs contribute to the structure of membrane lipid bilayers and lipid rafts, which play regulatory roles in cellular functions [40]. Sphingolipids (SPs) are crucial for brain function and development, serving as key structural components of plasma membranes. They regulate various cellular events by forming microdomains in the membrane and are involved in neuronal differentiation, synaptic transmission, and myelin stability. SPs enhance the functioning of ion channels and neuronal surface receptors, thereby influencing neuronal activity and gene expression [41].

**Cholesterol.** Cholesterol is a significant lipid class in the human body. It is the primary component of mammalian cell membranes and a fundamental functional unit. The brain has the highest cholesterol concentration, accounting for approximately 25% of the total cholesterol in the human body. Since cholesterol has difficulty crossing the BBB, it is mainly synthesized internally [6].

Myelin is the main storage form of cholesterol in the brain, containing nearly 80% of adult brain cholesterol [42]. It is crucial for myelin in both the central (CNS) and peripheral (PNS) nervous systems, synthesized by oligodendrocytes and Schwann cells, respectively [41]. Cholesterol is vital for neurotransmission and synaptic plasticity [43]. It supports cell membranes and myelin formation, contributing to the nervous system’s integrity. ApoE mediates the efficient recycling of cholesterol between neurons and glial cells [7]. Additionally, cholesterol is a precursor for steroid hormones and neurosteroids [4,6], underscoring its essential role in brain function.

### 2.2. Lipid Rafts

The complexity and diversity of lipids contribute to the heterogeneity and dynamic nature of neural lipid rafts, which are involved in various cellular processes in the nervous system. Lipid rafts are compact, dynamic zones within the membranes of brain cells, marked by gathering particular lipids like sphingolipids and cholesterol. These rafts are in different types of brain cells, including neurons, astrocytes, and microglia, and their lipid makeup is tailored to the specific requirements of each cell type [44]. Lipid rafts are essential for maintaining membrane structure and function, regulating multiprotein complexes involved in signal transduction and preserving brain homeostasis [44]. They also play a role in apoptosis and axonal growth in both the developing and adult central nervous system (CNS) [45].

### 2.3. Lipid Droplets

Lipid droplets primarily consist of cholesterol esters and triglycerides, featuring a hydrophobic core of neutral lipids surrounded by a phospholipid monolayer with proteins that regulate their function [17]. Their composition includes glycerolipids such as triacylglycerols (TAGs), diacylglycerols (DAGs), monoacylglycerols (MAGs), and cholesteryl esters (CEs), varying by cell type [42]. Key enzymes like diacylglycerol acyltransferases (DGAT1 and DGAT2) in the endoplasmic reticulum are essential for triglyceride synthesis and are important targets for modifying lipid droplet formation, especially in the nervous system [43,44,45,46,47,48,49,50].

The regulation of lipid droplet turnover involves two primary mechanisms: lipolysis and lipophagy. Lipolysis is triggered by hormonal signals like adrenaline and glucagon which activate lipase. The lipase, in turn, breaks down triglycerides into fatty acids and glycerol. Adipose triglyceride lipase (ATGL) is an important enzyme for breaking down triglycerides in the body, while in the brain, the primary enzyme responsible for this process is DDHD2, a gene that encodes an intracellular phospholipase A1, which is involved in lipid metabolism [51,52,53,54]. Lipophagy is a mechanism in which parts of fat droplets are engulfed by autophagosomes and combined to break down triglycerides. This process requires specific proteins and lysosomal enzymes and can be stimulated by factors like nutrient deficiency, cell stress, and certain signaling pathways, such as mTOR. Unlike lipolysis, lipophagy involves the internalization of fat droplets before their breakdown [48,55,56,57,58]. Although changes in lipid balance have been extensively studied in various neurological conditions, the focus on the emergence and role of lipid droplets in the brain is a relatively recent development. It is recognized that lipid droplets exist in nervous system cells throughout early development, aging, and neurological disorders, indicating the need for further investigation [17].

## 3. Brain Lipid Alterations in Aging: Implications for Structure and Function

The nervous system undergoes alterations in lipid homeostasis during physiological aging [4].

### 3.1. How Do Lipids Age?

Aging in the brain results in structural and cellular changes, including myelin degeneration and modifications in the neuronal extracellular matrix and perineuronal networks [59,60]. Functional decline mainly stems from dendrite loss, reduced spine density, and changes in trophic factor release [61]. Unlike other organs, brain aging selectively affects vulnerable neuronal populations [4,62]. Most lipids in the human brain have been shown to decrease in concentration after the age of 50. For example, the proportion of dry matter in the human brain decreases as nerve cell membrane lipids decline and the total myelin content reduces [63].

Furthermore, recent investigations have confirmed changes in lipid composition across different regions of the human brain during the aging process.

Burger and Seidel [64] and Rouser and Yamamoto [65] first linked aging to changes in the brain’s lipid profile using spectrophotometric analysis and chromatography. They noted that total lipid levels rise in the first two decades of life and then decline into adulthood and old age. Rouser and Yamamoto also identified a curvilinear relationship between lipid levels and age in the human brain. However, both studies examined the brain tissue as a whole, failing to differentiate between grey and white matter. This approach may overlook important differences that could impact various central nervous system regions [65].

Another crucial factor is the potential variability in lipid alterations affecting different brain regions. Subsequent examinations of various human brain areas have confirmed the presence of age-related lipid changes [66,67].

**Glycerolipids and Fatty Acids.** Glycerolipids and fatty acids are widely distributed in tissues and body fluids and are particularly abundant in the CNS, including various brain regions [68]. Higher levels of triacylglycerols (TAGs) may be associated with a decline in cognition in older adults [69]. The researchers found that higher triglyceride levels were associated with a slower decline in composite cognition, including global function, psychomotor speed, language, executive function, and memory [69]. However, the relationship between TAG and cognition is not entirely clear.

Studies indicate that both high and low TAG levels may negatively impact cognition and contribute to cognitive decline in aging, affecting neurotransmitter release and tau protein phosphorylation linked to AD [70,71]. TAG levels typically rise with age due to slower clearance, while monoacylglycerols (MAGs) and diacylglycerols (DAGs) significantly decrease in the amygdala of aged mice, impacting emotional memory and motivation [71].

Aging affects fatty acid levels and metabolism in the brain, especially omega-3 fatty acids like docosahexaenoic acid (DHA) and eicosapentaenoic acid (EPA). These fatty acids possess neuroprotective properties that may help mitigate cognitive decline related to aging [18].

**Glycerophospholipids.** As the brain ages, there is a notable reduction in glycerophospholipid proportions, with an approximately 10% loss of phosphatidylinositol (PI), phosphatidylethanolamine (PE), and phosphatidylcholine (PC) between the ages of 40 and 100 [9]. Research indicates that individuals aged 89–92 exhibit a 10 to 20% decrease in total phospholipids compared to controls aged 33–36, although some brain regions show unchanged phospholipid composition. The decline in phospholipid levels begins around age 20 and becomes more pronounced by age 80, with no significant gender differences. This highlights the progressive loss of lipid classes in the brain’s lipid matrix, particularly after age 50 [4,8].

**Sphingolipids.** Among sphingolipids, significant signaling changes during aging are those of gangliosides (GGs), a distinctive class of sphingolipids characterized by a complex structure [72]. They are widely distributed in tissues and body fluids and are particularly abundant in the CNS, including various brain regions [73]. Aging causes a decline in neural glycolipids, particularly gangliosides such as disialic acid type 1a (GD1a) and monosialic acid type 1 (GM1), with lesser declines in tri-sialic acid type 1b (GT1b) and di-sialic acid type 1b (GD1b) levels in the human frontal cortex. Similar findings have been noted in the rat cerebellar cortex, both in vivo and in vitro [74]. However, in the human hippocampus, only a moderate decrease in GD1a is observed, with no significant changes in other ganglioside classes. The visual cortex shows similar results, with minimal changes in individual ganglioside species [73].

**Cholesterol**. Scientific studies have shown that region-specific cholesterol shifts manifest as decreased cholesterol levels in human frontal and temporal cortices, the hippocampus, the caudate nucleus, and the cerebellum [8,9]. Laboratory models show a progressive reduction in cholesterol content across various brain regions, including the human cortex, hippocampus, cerebellum, mouse synaptosomes, and cultured hippocampal cells [8,9,14]. Cholesterol deficiency also increases the vulnerability of hippocampal glia to glutamate-induced excitotoxicity in primary cultures [4]. Aging-related cholesterol reduction impacts the synthesis and concentration of neurosteroids like pregnenolone (PREG) and dehydroepiandrosterone (DHEA), which are produced in the central and peripheral nervous systems, mainly by glial cells involved in myelin production using cholesterol or steroid precursors from other body regions [74].

Thus, changes in cholesterol and other lipid levels in nerve cells, such as lipid droplets [17], can have an impact on their structure and function. These changes can also affect the behavior of membrane-related proteins and signaling transduction by altering the fluidity of the neuronal membranes and, particularly, the lipid rafts, which are critical for signaling proteins [8]. While it is unclear if these changes are neutral or beneficial, the literature suggests that specific alterations in lipid patterns can affect biochemical properties, causing rearrangements in integrated proteins and contributing to neurodegenerative processes [20,75].

### 3.2. Lipid Rafts in Aging: The Anti-Inflammatory Role of A–I Binding Protein (AIBP)

Studies in humans have shown significant changes in lipid composition in lipid rafts, including specific classes of lipids such as PUFAs, plasmalogens, sphingomyelin, cholesterol, and sterol esters, as well as phospholipid-bound fatty acids like docosahexaenoic acid (DHA) and arachidonic acid (AA), related to brain aging [76].

There is evidence of a decrease in n-3 and n-6 PUFA during physiological aging in various areas of the human brain, especially the cortex and hippocampus [32]. It is noted that the proportions of different lipid classes gradually decrease with age; by 70, there is an increase in saturated fatty acids, while the main n-6 and n-3 PUFAs (AA and DHA) decrease with age, typically around 80 [77].

However, a recent study found no significant changes in the molar percentages of total lipid classes and fatty acids in the normal human frontal cortex throughout the lifespan, ranging from 24 to 85 years [14].

In addition to lipid reduction with age, molecular modifications of lipids, including oxidation, phosphorylation, and hydrolysis, are observed, leading to alterations and instability in lipid rafts. This affects ligand recognition by raft receptors due to altered raft structure and reduced receptor affinity, ultimately impacting brain function [78]. This condition can disrupt normal cellular processes and contribute to inflammatory conditions involving several inflammatory cells, causing oxidative stress [79]. Oxidative stress can hinder cholesterol efflux mechanisms in microglia, resulting in an increase in lipid rafts [80]. In this context, apolipoprotein A-I binding protein (AIBP) plays an important role. AIBP is a protein found in mice and humans, especially in the brain, kidney, reproductive organs, and liver. It helps regulate cholesterol movement out of cells, including endothelial cells, macrophages, and microglia and also affects inflammation by altering lipid rafts [81]. AIBP binds to Toll-like receptor 4 (TLR4) [80,82], which is important in the inflammatory response. By binding to TLR4, AIBP helps reduce inflammation and the activity of pro-inflammatory signals like NFκB and various cytokines [80,82,83]. When AIBP is deficient, it leads to increased inflammation and worsens conditions like atherosclerosis in animal models [82].

### 3.3. Lipid Droplets in Aging: The Controversial Role of Accumulation in the Microglia

During aging, lipid droplets tend to accumulate, playing a complex role that changes over the years. On one hand, lipid droplets are essential for preserving cellular integrity, as they help diminish the harmful effects of lipid peroxidation and protein aggregates. However, as people grow older, the excessive accumulation of these droplets can create issues within cells and tissues. Increased levels of extracellular lipids can lead to lipid droplet formation in microglia [84]. Microglia and astrocytes are cell populations that often operate in concert, but in certain contexts, such as inflammation, microglia behave autonomously [85].

Lipid droplet accumulation in microglia, named “lipid-droplet-accumulating microglia (LDAM)”, can be caused by various factors, including elevated concentrations of extra-cellular lipids, inflammatory events and chronic neuroinflammation, increased ROS levels, intracellular metabolic changes [84].

The accumulation of LDAMs leads to various consequences, resulting in cellular dysfunction and toxicity. Notably, LDAMs exhibit a significant phagocytosis deficit compared to microglia without lipid droplets, have numerous lysosomes that may impair the degradation of phagocytosed material, and show increased levels of intracellular ROS [46,86]. In the aging brain, these factors can disrupt dendritic arborization, causing neuronal damage and impaired activity. Additionally, high ROS levels can contribute to neuroinflammation, protein misfolding, and aggregation, potentially initiating age-related neurodegenerative diseases and cognitive disorders [87,88]. Nevertheless, while some studies suggest that LDAM may have detrimental effects, other research indicates that its role could be protective, depending on the context and cell type [88]. At present, the exact impact of LDAM on the aging brain, whether beneficial or detrimental, remains a subject of ongoing scientific studies.

Some studies suggest that LDAMs facilitate the synthesis of inflammatory signaling molecules, aiding both inflammation and neuroprotection in the brain [46]. In contrast, other research links LDAMs to a dysfunctional, proinflammatory state in the aging brain, marked by impaired phagocytosis and increased proinflammatory cytokine secretion [89]. Overall, lipid droplets provide protection against harmful peroxidized lipids in the brain [46] (Figure 1).

Other important storage organelles are peroxisomes, which are crucial for cellular functions, particularly in aging cells, as they break down fatty acids and manage hydrogen peroxide. Their efficiency declines with age, leading to metabolic issues and oxidative stress [90,91]. Changes in pathways like the peroxisome proliferator-activated receptor (PPAR) further worsen these problems. Additionally, dysfunctional peroxisomes disrupt cellular cleanup and mitochondrial interactions, contributing to aging and disease [92,93].

## 4. Brain Lipid Dysregulation in Alzheimer’s Disease: Mechanisms and Implications

Alzheimer’s disease (AD) is a progressive neurodegenerative condition that accounts for approximately 70% of all cases of dementia worldwide. This pathology causes irreversible neurological decline, characterized by cognitive and behavioral deterioration. Its complex pathogenesis involves genetic, environmental, vascular, metabolic, and energetic factors [94,95]. The Aβ peptide and tau protein are key agents in AD, forming amyloid plaques and neurofibrillary tangles. The development of senile plaques is linked to astrogliosis [96], neuroinflammation [77], and oxidative stress [97], particularly affecting the neocortex and hippocampus, which are the most vulnerable brain regions [95].

Although the exact molecular mechanisms are unclear, studies show that neurological disorders, especially AD, involve high lipid content and lipid dyshomeostasis, which significantly contribute to disease development [96]. While abnormal lipid accumulation was recognized in early AD neuropathology, recent research has focused on lipid dysregulation. Lipidomic and metabolomic studies reveal alterations in various lipid classes throughout different phases of AD, indicating that dysregulation, particularly of sphingolipids, cholesterol and other lipid classes, plays a crucial role in AD onset and progression [96] (Figure 3).

A recent study by Yin’s group [96] found that metabolically deficient astrocytes can induce neuroinflammation and neurodegeneration. Impaired lipid waste removal by mitochondria in these cells leads to lipid accumulation in the brain, transforming astrocytes into pro-inflammatory and neurotoxic cells, which are early hallmarks of AD. The same group also showed that the APOE4 gene, a major AD risk factor, suppresses lipid degradation in astrocytes, linking impaired lipid clearance to neurodegeneration [97]. While ApoE4 increases amyloid-β (Aβ) production, it also regulates Aβ clearance [98]. Besides the APOE gene, numerous genes directly involved in lipid metabolism, such as CLU (clusterin, also known as apolipoprotein J), SORL1 (sortilin-related receptor 1), and ABCA7 (ATP-binding cassette, sub-family A, member 7), have been associated with AD [99].

Lipid alterations vary at different stages of AD. Early AD is characterized by dysregulated lipid metabolism in the cortex, involving pathways like PPAR signaling, glycerophospholipid metabolism, and fatty acid biosynthesis [100]. These disturbances are linked to lipid peroxidation, oxidative stress, mitochondrial dysfunction, and neuronal death, particularly emphasizing sphingolipid metabolism’s role in AD progression [94]. Lipid peroxidation, driven by oxidative stress and ROS targeting PUFAs, is an early event in AD [101,102]. Increased activity of lipoxygenases (LOXs) catalyzes the oxidation of polyunsaturated fatty acids, leading to elevated lipid peroxidation products. Other enzymes, such as cyclooxygenases (COXs) and NADPH oxidases (NOXs), may also contribute to neurodegeneration [103,104,105]. Further research has revealed intricate interactions between lipid metabolism and key pathogenic mechanisms in AD, including amyloid genesis, bioenergetic deficits, oxidative stress, neuroinflammation, and myelin degeneration [94].

### 4.1. How Do Lipids Behave in Alzheimer’s Disease?

Several lipid classes undergo morphological and functional alterations in patients suffering from AD. Specific lipids affected include glycerolipids, glycerophospholipids, sphingolipids, and cholesterol (Figure 2).

**Figure 2 biomolecules-14-01362-f002:**
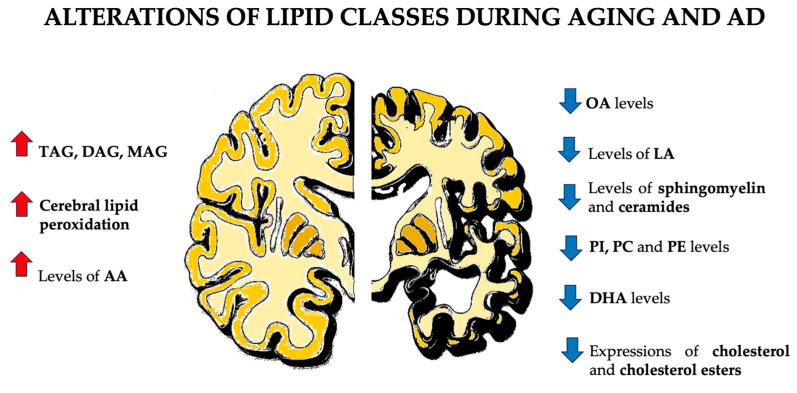
Several lipid classes undergo morphological and functional alterations in patients suffering from AD. Abbreviations: AA: arachidonic acid; MAG: monoacylglycerol; DAG: diacylglycerol; TAG: triacylglycerols, OA: oleic acid; LA: linoleic acid; PC: phosphatidylcholine; PE: phosphatidylethanolamine; DHA: docosahexaenoic acid.

**Glycerolipids and Fatty Acids**. Glycerolipids play a significant role in the pathogenesis of AD. Their reduction in AD seems more closely linked to changes in metabolic pathways rather than neuronal death [106].

In fact, several studies have shown that glycerolipid metabolism appears to be altered in AD, as one of the early biochemical changes in AD pathogenesis [107]. In AD, dysfunctions in enzymes involved in TAG biogenesis, such as diacylglycerol acyl-transferases (DGATs), can lead to abnormal TAG accumulation, which may contribute to neurotoxicity and synaptic dysfunction [26].

Triacylglycerols (TAGs) are elevated in the brains of AD patients and animal models, with a 14-fold increase in specific regions compared to healthy controls [108]. However, while higher brain TAG levels are linked to AD, elevated circulating TAG levels are associated with a lower risk of dementia in older adults [109]. Other glycerolipid species, such as monoacylglycerols (MAGs) and diacylglycerols (DAGs), are also dysregulated in AD [18]. Increased MAG levels may relate to monoacylglycerol lipase (MAGL) dysregulation [110]. Inhibiting MAGL in AD models reduces Aβ plaque accumulation and has neuroprotective effects [38,111,112]. Elevated DAG levels may indicate enzyme overactivation, disrupting cellular functions [113]. High MAG and DAG levels are present in the frontal cortex and blood of AD patients even in early stages [32,33], suggesting these alterations signal early biochemical changes in AD. However, DAG levels remain elevated also in AD progression [34,114]. Fatty acids (FAs) also influence AD development, with some promoting plaque formation and others preventing amyloid fibril formation [115,116,117]. Altered FA levels are found in vulnerable brain regions of AD patients [118].

**Glycerophospholipids.** Alterations in the levels of glycerophospholipids, such as phosphatidylcholines (PCs) and phosphatidylethanolamines (PEs), have been observed in AD and could indeed be associated with various processes [113]. A portion of glycerophospholipid reduction is linked to neuronal loss, as these lipids are critical components of cell membranes. As neurons die, the overall demand for these lipids decreases, contributing to their reduction in AD. However, metabolic dysregulation strongly influences glycerophospholipid depletion, particularly in pathways that regulate synaptic function and neurotransmission. This includes impairments in the cholinergic system, independent of cellular loss [119]. Oxidative stress, a hallmark of AD, leads to the peroxidation of unsaturated fatty acids in glycerophospholipids, generating products like hydroxyeicosatetraenoic acids (HETEs) and isoprostanes [120]. This process alters cell membrane integrity, impacting neuronal signaling and synaptic function. Dysregulation of phospholipase A2 (PLA2) activity has also been reported in AD [121]. Increased PLA2 activity hydrolyzes glycerophospholipids into lysophospholipids and free fatty acids, contributing to oxidative stress, inflammation, and neuronal dysfunction.

Changes in glycerophospholipid synthesis may further explain the altered lipid levels in AD. Dysregulation of enzymes involved in phospholipid biosynthesis pathways, such as phosphatidylcholine synthase and phosphatidylethanolamine methyltransferase, could lead to imbalances in phospholipid composition. Changes in lipid metabolism pathways, including those regulated by enzymes such as fatty acid desaturases and acyltransferases, may also influence the synthesis and turnover of glycerophospholipids in AD [122]. Therefore, alterations in glycerophospholipid levels observed in AD could result from a combination of factors, including oxidative-stress-mediated lipid peroxidation, dysregulation of phospholipase activity, and changes in phospholipid synthesis pathways.

In early AD, glycerophospholipid levels in grey and white matter remain relatively stable in the frontal cortex and cerebellum, suggesting minimal changes initially [113]. As the disease progresses, there is a decline in glycerophospholipid concentrations, such as phosphatidylinositol (PI), phosphatidylethanolamine (PE), and phosphatidylcholine (PC) [122]. In APPJ20 mouse models, total PE and PC increased at 3 months but decreased at 6 months in the cortex compared to wild-type mice, indicating time- and area-dependent changes in phospholipid composition that are important for understanding AD pathology [123]. 

**Sphingolipids.** Sphingomyelin (SM) and ceramide are the primary sphingolipids altered in AD brains [4]. Sphingolipid metabolism is disrupted in AD, with elevated ceramide levels contributing to neuroinflammation and cell death [113]. This dysregulation suggests a metabolic imbalance that accelerates neuronal damage. In the early stages of AD, SM levels show inconsistent results; some studies report decreased levels in soluble fractions while remaining unchanged in membrane fractions [124]. Increased SM is found in the cerebellum but not in the frontal cortex of AD patients [124]. Conversely, ceramide concentrations and sphingolipid-derived molecules increase, attributed to elevated sphingomyelinase (SMase) activity [125], which depletes SM and promotes abnormal APP processing [126]. Experimental evidence indicates that the upregulation of SM synthase (SMSs) contributes to Aβ plaque formation, neuroinflammation, and cognitive decline in AD mice [38,127,128]. While ceramide levels rise in early AD, they may decrease in later stages [125]. Importantly, alterations in other sphingolipids may also significantly impact AD pathology. Broader sphingolipid profiles in AD reveal complex changes beyond SM and ceramide, including reductions in sulfatides and other metabolites like sphingosine-1-phosphate [129].

Sulfatides, a type of glycosphingolipid sulfate, show significant depletion in both gray and white matter as AD progresses [130]. Studies indicate a notable decrease in sulfatide levels in the cerebrospinal fluid of AD patients compared to healthy controls, with significant reductions in brain regions involved in pathology, such as the hippocampus and cerebral cortex [129,131]. Additionally, specific changes in ganglioside composition include decreases in major gangliosides (GM1, GD1a, GD1b, and GT1b) and increases in simpler gangliosides (GM2, GM3, and GD3) in AD brains [18,73]. Research has shown a decline in total sialic acid levels and a reduction in a-series gangliosides like GM1 and GD1a in AD cases compared to controls. Notably, there is an increase in Chol-1α gangliosides and GT1aα and GQ1bα, specific markers associated with cholinergic neurons in the brains of AD patients [132].

**Cholesterol.** Dysregulation of cholesterol metabolism in the brain significantly contributes to the development and progression of AD, though its influence remains controversial. Cholesterol changes are mainly linked to metabolic alterations rather than neuronal loss [106]. While peripheral cholesterol levels minimally impact AD due to the blood–brain barrier (BBB), brain cholesterol levels warrant further investigation [132,133,134]. Excessive cholesterol accumulation in the brain correlates with an increased risk of AD [133,134,135]. Notably, total cholesterol levels increase in brain regions with extensive Aβ deposits and NFTs, particularly the entorhinal cortex [94].

Cholesterol plays a crucial role in forming Aβ plaques and regulates their fibrillation, transport, and clearance. Elevated membrane cholesterol levels lead to Aβ incorporation into membranes and increased cytosolic calcium in astrocytes, contributing to neuronal cell death [28,135,136].

Additionally, oxysterols like 24-hydroxycholesterol (24-OH Chol) and 27-hydroxycholesterol (27-OH Chol) are linked to AD progression, enhancing BACE1 activity and regulating Aβ aggregation and misfolding [136,137,138].

**Figure 3 biomolecules-14-01362-f003:**
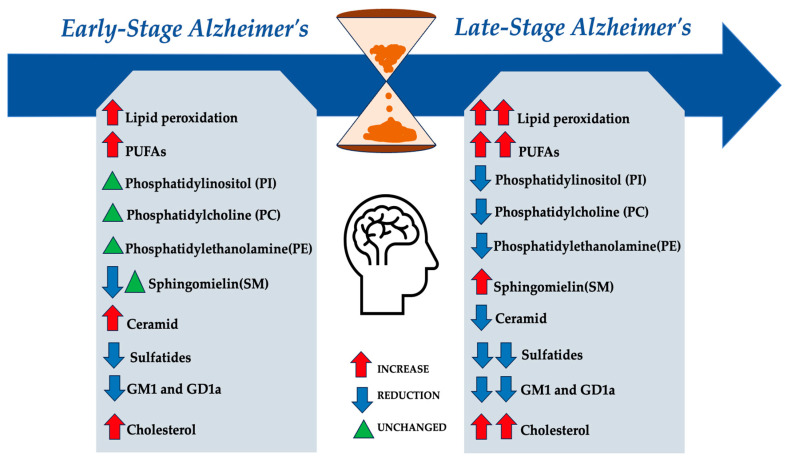
Different alterations of the various lipid classes in the brain during the early stage and late stages of Alzheimer’s disease [108,110,113,115,124,131,132,133,134,135].

### 4.2. Lipid Rafts

In the AD brain, lipid rafts, specialized microdomains in cell membranes rich in cholesterol and sphingolipids, play a crucial role in the pathogenesis of the disease.

Numerous AD-associated proteins, such as β-amyloid precursor protein (βAPP), β-secretase, γ -secretase, and neprilysin, have also been found in lipid rafts [139], contributing to the generation of amyloid-beta(Aβ) peptides, and to the formation of amyloid plaques, a hallmark of AD. Therefore, lipid rafts represent the site where Aβ interacts with ApoE and tau, favoring the aggregation of Aβ oligomers and hyperphosphorylated tau [19]. The process of amyloid genesis becomes, therefore, closely related to the lipid composition in lipid rafts [140].

Additionally, alterations in lipid rafts can affect the function of various mem-brane-bound proteins involved in synaptic transmission and neuronal signaling, further exacerbating the neurodegenerative process in AD.

In early AD forms, FA alterations in lipid rafts are already apparent, especially at the frontal and entorhinal cortex levels [141]. Lipid rafts extracted from the frontal cortex of individuals with AD show diminished levels of n-3 long-chain polyunsaturated fatty acids (LC-PUFAs), particularly docosahexaenoic acid (DHA), as well as monoenes, mainly oleic acid. Furthermore, both the unsaturation and peroxidability indices are significantly lower when compared to those of healthy controls [14].

Lipids like GM1 ganglioside, sphingomyelin, and cholesterol promote the formation of cytotoxic Aβ fibrils [142]. Aβ oligomers can disrupt cholesterol levels in lipid microdomains, showing an inverse relationship between cholesterol and Aβ. Changes in cholesterol levels within lipid rafts significantly impact their properties and play a crucial role in processing amyloid precursor protein (APP) and producing amyloid-beta (Aβ) [4,39,40,143,144]. Cholesterol is crucial for the incorporation of amyloid precursor protein (APP) into lipid rafts, aiding its movement into these specialized microdomains [145]. Wahrle et al. [40] found that the activity of γ-secretase can be influenced by changes in cholesterol levels at the cell membrane. The activity of γ-secretase is inhibited when cholesterol levels decrease while restoring cholesterol stores renews this protein’s function (Figure 4A).

The relationship between cholesterol levels in lipid rafts and γ-secretase activity also promotes the cleavage of APP, which is influenced by the presence of BACE1, ultimately leading to increased Aβ production within these rafts. BACE1, located in lipid rafts, is stabilized by three palmitoylated residues in its structure [146,147].

### 4.3. Lipid Droplets

The buildup of lipid droplets has emerged as a crucial factor in AD, particularly after the establishment of Aβ plaques and neurofibrillary tangles. This accumulation is notably observed in models with diverse genetic backgrounds [20].

A key mechanism for glial lipid droplet formation in AD involves toxic lipids from neurons [88,148], facilitated by ApoE, a genetic risk factor for late-onset AD. The ApoE4 allele is associated with the highest risk, while the E2 and E3 alleles provide neuroprotective effects. The isoforms of ApoE differ in lipid binding and stability, with ApoE4 being less stable and having a lower affinity for brain-specific lipoprotein particles [17,149,150].

In Drosophila models, knocking down the apolipoprotein lazarillo completely inhibits glial lipid droplet formation during oxidative stress, an effect restored by human ApoE3 but not ApoE4 [144]. In mammalian cells, reducing ApoE levels in either neurons or glia diminishes lipid transport and lipid droplet formation [97,148,151].

Neurons produce ApoE under oxidative stress, suggesting its increase enhances lipid transfer [151]. However, the precise mechanisms by which ApoE from different cell types contributes to lipid transport are not fully understood. ApoE from astrocytes likely aids in transporting lipoprotein particles rich in neuronal lipids for endocytosis by astrocytes.

This lipid transport mechanism is supported by several genetic factors linked to AD risk [70].

ApoE significantly influences the quantity of lipid droplets in cells. In the absence of neurons, glial cells expressing ApoE4 tend to accumulate more lipid droplets, highlighting the bidirectional nature of lipid transport in the brain.

Without functional ApoE, the transfer of lipoprotein-like particles from astrocytes to neurons decreases, causing neutral lipid buildup in astrocytes, while stressed neurons struggle to transport excess toxic lipids to astrocytes during oxidative stress. In both scenarios, glial lipid droplets play a neuroprotective role; their failure to form can lead to increased neurodegeneration [148,152] (Figure 4B).

**Figure 4 biomolecules-14-01362-f004:**
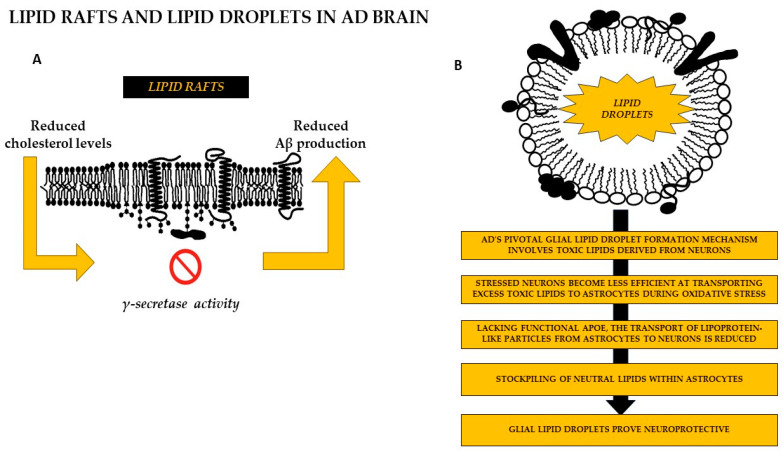
The role of cholesterol from lipid rafts and lipid droplets in AD brain. (**A**) Cholesterol, a key player in integrating APP within lipid rafts, facilitates its trafficking into these microdomains. Wahrle et al. reported that γ -secretase activity, could be modulated by fluctuations in cholesterol levels at the cell membrane. The activity of γ-secretase is inhibited when cholesterol levels decrease while restoring cholesterol stores renews this protein’s function [39,40,143,144]. This figure is licensed under the Creative Commons Attribution-Share Alike 2.5. (**B**) Lipid droplet accumulation becomes prominent in AD pathology, emphasizing the intricate link between lipid homeostasis, metabolism, and brain aging. AD’s pivotal glial lipid droplet formation mechanism involves toxic lipids derived from neurons. Their role as illustrated in the figure would be neuroprotective [151,152,153,154,155,156]. Abbreviations: AD: Alzheimer’s disease; ApoE: Apolipoprotein E.

Besides regulating lipid transport, ApoE likely influences lipid droplet dynamics through direct and indirect modulation of lipid metabolism and membrane trafficking [153,154,155]. Specifically, the APOE4 isoform is linked to increased lipid droplet accumulation in microglia, with a negative correlation to cognitive performance and a positive correlation to Aβ plaque burden and tau pathology in the brains of AD patients [156]. Peroxisomes also contribute to AD by impacting Aβ clearance and exacerbating oxidative stress, neuronal dysfunction, and neuroinflammation. Dysfunctional peroxisomes can impair the synthesis of specialized lipid mediators, further worsening neuroinflammation and neuronal damage [157,158]. Lipid homeostasis is crucial for the health of the nervous system.

While changes in lipid homeostasis are well documented in various neuropathologies, the study of lipid droplets in the brain has gained attention more recently. It is now recognized that lipid droplets are present in nervous system cells during early development, aging, and neuropathologies, suggesting new research opportunities [17]. Alterations in lipid droplets, along with changes in lipid rafts and various lipid classes, are summarized in Table 1.

## 5. Conclusions

The brain relies on lipids for essential processes such as neurogenesis, signal transduction, and modulation of gene expression. Thus, age-related changes in lipid metabolism can significantly impact brain function, particularly in neurodegenerative diseases like AD. As people live longer, the prevalence of dementia, including AD, is increasing, presenting significant challenges for public health and the economy [159]. Developing improved methods for early diagnosis and monitoring of these conditions is imperative. Currently, specialists primarily rely on tests and brain scans for AD diagnosis, but they struggle to identify the disease in its early stages when symptoms are not yet present [160,161].

However, changes in lipid metabolism, have been linked to AD, making these lipid changes potential clinical biomarkers for early diagnosis across various human samples, including blood, plasma, serum, urine, and cerebrospinal fluid (CSF) [162,163,164,165,166,167,168,169,170,171].

In particular, the reported alterations in membrane lipid composition and lipid homeostasis directly affect brain health, influencing lipid rafts and the function of associated proteins involved in neurodegenerative processes.

Also, lipid droplets emerge as key players in the nervous system, particularly concerning aging and neurodegenerative diseases. Their role in regulating cellular stress has gained attention, providing insights into their protective function against lipid-mediated toxicity. Understanding how lipid droplets sequester fatty acids during cellular stress is crucial for unraveling their contributions to brain health, especially in AD.

In this regard, lipidomics shows great promise for identifying clinical markers [162]. Lipidomics, through its ability to study changes in lipid classes and associated proteins, advances our understanding of AD, offering new clinical markers for early diagnosis and insights into disease mechanisms [172,173].

Further studies are essential to elucidate the connections between lipid metabolism, changes in lipid rafts, and lipid droplet accumulation in the context of aging and AD.

This knowledge can pave the way for early diagnosis and potential strategies to slow disease progression.

## Figures and Tables

**Figure 1 biomolecules-14-01362-f001:**
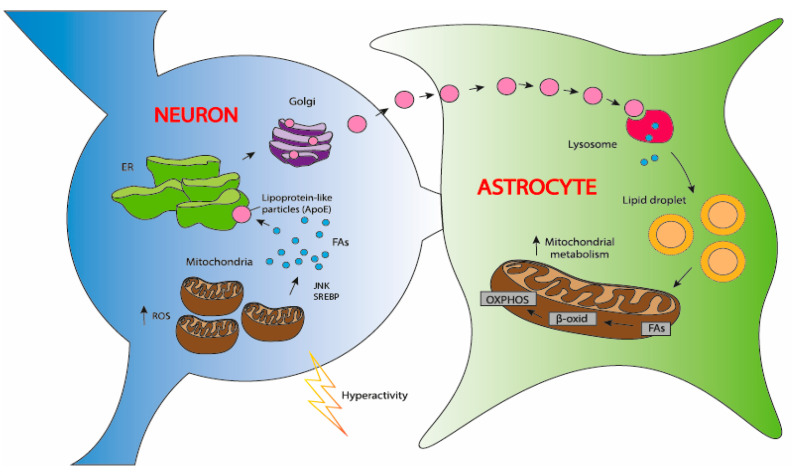
Neuroprotective role of lipid droplets (LDs) in the nervous system in the aging brain. In the aging brain, defective mitochondria in neurons generate ROS, activating the JNK and SREBP transcription factors that stimulate lipid synthesis. These newly synthesized lipids undergo peroxidation in the presence of ROS and are subsequently transported to glia, where they are stored in lipid droplets. This process is facilitated by the apolipoprotein E (ApoE). ApoE is primarily produced by astrocytes in the brain and plays a critical role in maintaining and repairing neurons. One of its essential functions is transporting cholesterol and other lipids to neurons, supporting their growth and function. When ROS levels are elevated, the neuroprotective role of glial LDs becomes evident [46,88]. Abbreviations: ROS: reactive oxygen species; FAs: fatty acids; ApoE: Apolipoprotein E; OXPHOS: oxidative phosphorylation; β-oxid: β-oxidation; JNK: c-Jun N-terminal kinase; SREBP: Sterol regulatory element-binding protein.

**Table 1 biomolecules-14-01362-t001:** Summary of lipid alterations in AD.

Lipid Class	Key Findings	Increase/ Decrease
**Glycerolipids and Fatty** **Acids**	- **Triacylglycerols (TAGs):** Elevated levels in AD brains, particularly in regions like the ependymal cell layer and choroid plexus. - **Diacylglycerols (DAGs):** Accumulation linked to decreased conversion to phosphatidic acid and overactivation of phospholipases. - **Monoacylglycerols (MAGs):** Dysregulated due to monoacylglycerol lipase (MAGL) inactivation leading to reduced Aβ production.	TAG: **Increase** DAG: **Increase** MAG: **Increase**
**Glycero** **phospholipids**	- **Phosphatidylcholines (PCs):** Decreased levels associated with neuronal loss and metabolic dysregulation. - **Phosphatidylethanolamines (PEs):** Show a decrease as AD progresses. Early stages show relatively stable levels in grey and white matter. - **Phosphatidylinositols (PIs):** As AD levels decrease, they disrupt key signaling pathways like IP3 and PI3K. This reduction contributes to synaptic dysfunction, neuroinflammation, and potentially the accumulation of beta-amyloid plaques, worsening cognitive decline and neuronal degeneration.	PC: **Decrease** PE: **Decrease** PI: **Decrease**
**Sphingolipids**	- **Sphingomyelin (SM):** Levels show variability; some studies indicate decreased levels in soluble fractions but unchanged in membrane fractions. - **Ceramides:** Increased levels contribute to neuroinflammation and cellular stress. - **Sulfatides:** Significant depletion in both grey and white matter as AD progresses. - **Gangliosides:** Decreased major gangliosides (GM1, GD1a) while simpler gangliosides (GM2, GM3) increase.	SM: **Variable** Ceramide: **Increase** Sulfatides: **Decrease** Gangliosides: **Altered**
**Cholesterol**	- **Total Cholesterol:** Increased levels in regions with extensive Aβ deposits, particularly in the entorhinal cortex. - **Oxysterols:** Elevated 24-hydroxycholesterol and 27-hydroxycholesterol implicated in Aβ aggregation and metabolism.	Total Cholesterol: **Increase** Oxysterols: **Increase**
**Lipid Rafts**	- **Composition Changes:** Reduced levels of n-3 LC-PUFAs (e.g., DHA) and monoenes (e.g., oleic acid) in lipid rafts from AD patients. - **Aβ Generation:** Lipid rafts rich in cholesterol and sphingolipids facilitate Aβ peptide production and aggregation.	PUFAs: **Decrease** Cholesterol: **Variable**
**Lipid Droplets**	- **Formation Mechanism:** Increased accumulation of lipid droplets in glial cells derived from neuronal toxic lipids. - **ApoE Role:** Impaired lipid transport related to ApoE genotype; ApoE4 allele leads to more lipid droplet accumulation.	Lipid Droplets: **Increase**

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
