# Peer review of "Exploring the Dynamic Changes of Brain Lipids, Lipid Rafts, and Lipid Droplets in Aging and Alzheimer’s Disease"

_biomolecules, 2024, doi:10.3390/biom14111362_

Round 1
Reviewer 1 Report
Comments and Suggestions for Authors
This review titled with “Exploring the Dynamic Changes of Brain Lipids, Lipid Rafts, and Lipid Droplets in Aging and Alzheimer’s Disease” by Michele Cerasuolo et al. They comprehensively examine the intricate alterations in lipid profiles across various brain regions as a result of aging, and their implications for neurodegenerative diseases such as Alzheimer’s disease (AD). The authors highlight that while the overall lipid profile in the human brain remains relatively stable throughout adulthood, notable changes begin to manifest post-50 years. These include a gradual decrease in total lipid content and alterations in the composition of fatty acids, glycerophospholipids, and cholesterol levels, which vary by brain region.
Reviewer Comments:
1. Length of Manuscript: The manuscript is currently too lengthy, making it challenging to engage with and retain the information provided.
2. Need for Visual Summaries: The second part of the manuscript would benefit greatly from the inclusion of summary figures or tables to better illustrate the complex data and findings discussed.
3. Chapter Titles: The titles of the chapters are too vague and need to be made more descriptive to accurately reflect the content and aid in navigation.
4. Textual Corrections: There are minor textual errors that need correction, such as the formatting mistake in Line 423 “85]. ].” These should be addressed to enhance the readability and professional quality of the manuscript.
To improve the manuscript, I recommend shortening the text where possible to enhance readability, incorporating summary figures or tables in the second part to visually represent key findings, refining chapter titles to be more indicative of the contents, and correcting all textual errors to maintain a professional standard.”
Author Response
Comments 1: Length of Manuscript: The manuscript is currently too lengthy, making it challenging to engage with and retain the information provided.
Response 1: Thank you for your valuable suggestion. In response, we have carefully reviewed and synthesized the manuscript, streamlining the content while ensuring that all key points and essential information are retained. The revisions have been made with attention to preserving the accuracy and integrity of the original submission.
Comments 2: Need for Visual Summaries: The second part of the manuscript would benefit greatly from the inclusion of summary figures or tables to better illustrate the complex data and findings discussed.
Response 2: Thank you for your insightful feedback. In response, we have compiled a comprehensive synoptic table that outlines the mechanisms of all lipid classes, including lipid rafts and lipid droplets, in the context of Alzheimer's disease. This table has been carefully designed to provide a clear and concise overview, integrating the relevant findings and mechanisms associated with these lipid structures and their roles in the disease pathology. (See line 830- Table 1 Summary of lipid Alteration in AD)
Comments 3: Chapter Titles: The titles of the chapters are too vague and need to be made more descriptive to accurately reflect the content and aid in navigation.
Response 3: We have made the titles of the various paragraphs more explanatory and elaborate. Thank you.
(See lines 102, 262, 463, 835)
Comments 4: Textual Corrections: There are minor textual errors that need correction, such as the formatting mistake in Line 423 “85]. ].” These should be addressed to enhance the readability and professional quality of the manuscript.
Response 4: We corrected minor textual errors. Thank you.
Reviewer 2 Report
Comments and Suggestions for Authors
This is a timely review on a relevant scientific topic. It is written clearly and concisely. This review has been carefully performed and I only have few suggestions of minor modifications to improve several points, see below.
1. A point of high scientific relevance is whether or not the decrease in the concentrations of lipid species in the human brain reported after the age of 50 years and in Alzheimer’s disease (AD) could be simply monitoring a decrease of brain cells or cellular membrane structures (like loss of neuronal extensions). Note that only in the cases where this correlation can be excluded the changes should be rationalized in terms of dysregulation of lipid metabolism. Since this is likely to vary between different lipid species, this point should be addressed for the different lipids analyzed in this review.
2. As the amyloid β oligomers have been shown to be the most potent neurotoxic species, it is recommended the modify the following sentence of lines 82-83 to avoid misinterpretations including the underlined word diagnosis: “It is known that AD is characterized by the presence of insoluble aggregates of amyloid β (Aβ), forming neuritic plaques (NPs), which play a central role in the diagnosis of disease’s pathology.”
3. The following sentence of lines 390-392 contains several very strong generic statements: “However, in addition to the reduction of lipid age related, molecular modifications of lipids are also observed during aging, causing in turn alterations and instability in lipid rafts, weakness in the recognition of ligands by raft receptors and changes in the membrane, ultimately impacting brain function[79].” Due to the relevance of two of the mentioned scientific points (molecular modifications of lipids, alterations of ligands recognition by raft receptors), it is recommended to include next several additional sentences providing a brief but more detailed description of these changes.
4. Please, rewrite the following sentence of lines 104-105 for the sake of clarity (underlined words): “The basic structure of lipids is modulated by the relative polarity of the hydrophobic codes concerning the head group[21].”
5. Please, revise the text for correction of typo errors like “progresse” (line 590-591), “In-Dicators” (heading of line 800), and “rROS” (line 789).
6. Please, normalize citations of references in the text. Note that in parts of the text citations are written as superscript characters within brackets, for example, in sections 1. Introduction, 3. Brain Lipids in Aging and, also, subsection dealing with Cholesterol.
Author Response
Comments 1: A point of high scientific relevance is whether or not the decrease in the concentrations of lipid species in the human brain reported after the age of 50 years and in Alzheimer’s disease (AD) could be simply monitoring a decrease of brain cells or cellular membrane structures (like loss of neuronal extensions). Note that only in the cases where this correlation can be excluded the changes should be rationalized in terms of dysregulation of lipid metabolism. Since this is likely to vary between different lipid species, this point should be addressed for the different lipids analyzed in this review.
Response 1: The decrease in lipid concentrations in the brain after age 50 and in Alzheimer's disease (AD) arises from both neuronal loss and metabolic dysregulation. Research indicates reductions in specific lipids, like phospholipids and sphingolipids, are associated with neuronal atrophy and synaptic loss, especially in later stages of AD. Additionally, lipidomic studies reveal significant disruptions in lipid metabolism independent of cellular loss, including alterations in glycerolipid and sphingolipid pathways. We have clarified these aspects for each lipid class, highlighting how both factors contribute to lipid changes in the AD brain, with varying impacts based on disease progression. (See lines 577-578; 613-619; 655-656; 692-693)
Comments 2: As the amyloid β oligomers have been shown to be the most potent neurotoxic species, it is recommended the modify the following sentence of lines 82-83 to avoid misinterpretations including the underlined word diagnosis: “It is known that AD is characterized by the presence of insoluble aggregates of amyloid β (Aβ), forming neuritic plaques (NPs), which play a central role in the diagnosis of disease’s pathology.”
Response 2: "The sentence in the text is as follows: " “It is known that AD is characterized by the presence of insoluble aggregates of amyloid β (Aβ), forming neuritic plaques (NPs), which play a central role in the disease’s pathology”. It never mentions the diagnosis of the disease but refers to its role in the pathogenesis of the disease.
Comments 3: The following sentence of lines 390-392 contains several very strong generic statements: “However, in addition to the reduction of lipid age related, molecular modifications of lipids are also observed during aging, causing in turn alterations and instability in lipid rafts, weakness in the recognition of ligands by raft receptors and changes in the membrane, ultimately impacting brain function[79].” Due to the relevance of two of the mentioned scientific points (molecular modifications of lipids, alterations of ligands recognition by raft receptors), it is recommended to include next several additional sentences providing a brief but more detailed description of these changes.
Response 3: Thank you for the thought-provoking insight. We have modified the paragraph as follows:
“However, in addition to the reduction of lipids related to age, molecular modifications of lipids, including oxidation, phosphorylation, and hydrolysis, are also observed during aging. This causes alterations and instability in lipid rafts, leading to weakness in the recognition of ligands by raft receptors due to altered raft structure and reduced receptor affinity, along with changes in the membrane, ultimately impacting brain function [79].” (See lines 371-376)
Comments 4: Please, rewrite the following sentence of lines 104-105 for the sake of clarity (underlined words): “The basic structure of lipids is modulated by the relative polarity of the hydrophobic codes concerning the head group[21].”
Response 4: We have restructured the paragraph to clarify the concept. Thank you.
"The fundamental structure of lipids is shaped by the differences in polarity between their hydrophobic tails and the hydrophilic head group. Specifically, the varying degrees of polarity in these components affect how lipids interact with each other and with their environment, ultimately influencing their functional roles in biological membranes."(See lines 107-111)
Comments 5: Please, revise the text for correction of typo errors like “progresse” (line 590-591), “In-Dicators” (heading of line 800), and “rROS” (line 789).
Response 5: We have corrected the typing errors. Thank you.
Comments 6: Please, normalize citations of references in the text. Note that in parts of the text citations are written as superscript characters within brackets, for example, in sections 1. Introduction, 3. Brain Lipids in Aging and, also, subsection dealing with Cholesterol.
Response 6: That's fine, thank you. We have normalized the citations of the references in the text.
Round 2
Reviewer 1 Report
Comments and Suggestions for Authors
From my perspective, the manuscript remains somewhat lengthy and could potentially deter reader engagement due to its protracted nature. It may benefit from further condensation to enhance its readability and appeal.
Author Response
Dear Reviewer
Thank you very much for taking the time to review this manuscript.
As required, we have reduced all the chapters in order to improve and facilitate readability while more text sections remained unchanged.
Therefore, we have proceeded to modify the following portions of the text.
The test lines of the previous manuscript and the corresponding replaced lines in the new manuscript version are reported below:
Lines 39-46 with lines 39-45
Lines 102-110 with lines 102-107
Lines 114-129 with lines 111-121
Lines 140-148 with lines 134-141
Lines 155-163- with lines 146-152
Lines167- 183 with lines 156-168
Lines 192-199 with lines 176-180
Lines 206-218 with lines 187-194
Lines 226-240 with lines 202-215
Lines 249-254 with lines 222-226
Lines 264-271 with lines 238-242
Lines 277-284 with lines 248-252
Lines 298- 321 with lines 265-283
Lines 325-333 with lines 287-294
Lines 336-345 with lines 297-305
Lines 350-397 with lines 310-342
Lines 411-420 with lines 356-363
Lines 425-431 with lines 368-373
Lines 453-459 with lines 396-401
Lines 462-481 with lines 406-420
Lines 503-516 with lines 441-447
Lines 521-539 with lines 452-461
Lines 581-606 with lines 498-511
Lines 617-629 with lines 520-528
Lines 639-710 with lines 537-587
Lines 730-736 with lines 608-613
Lines 748-778 with lines 624-650
Lines 810-826 with lines 687-702
Lines 831-887 with lines 726-752